Areas of spatial overlap between common bottlenose dolphin, recreational boating, and small-scale fishery: management insights from modelling exercises

La Manna Gabriella gabriella.lamanna@gmail.com 1 2 3
Ronchetti Fabio 2
Perretti Francesco 2
Ceccherelli Giulia 1 2
1 University of Sassari , Sassari , Italy
2 Environmental Research and Conservation, MareTerra Onlus , Alghero , Italy
3 National Biodiversity Future Centre , Palermo , Italy
Yapıcı Sercan
Electronic publication date: 2023 Sep 28
Publication date: 2023
Volume: 11
Electronic Location ID: e16111
Received 2023 May 19; Accepted 2023 Aug 27
Copyright: ©2023 La Manna et al.
Copyright year: 2023
Copyright holder: La Manna et al.
License: This is an open access article distributed under the terms of the Creative Commons Attribution License, which permits unrestricted use, distribution, reproduction and adaptation in any medium and for any purpose provided that it is properly attributed. For attribution, the original author(s), title, publication source (PeerJ) and either DOI or URL of the article must be cited.
License URL: https://creativecommons.org/licenses/by/4.0/

Keywords: Tursiops truncatus, Habitat use, Fishing activities, Boat traffic

Funding: The authors received no funding for this work.

==============================
Background

Sustainable management requires spatial mapping of both species distribution and human activities to identify potential risk of conflict. The common bottlenose dolphin (Tursiops truncatus) is a priority species of the European Union Habitat Directive, thus, to promote its conservation, the understanding of habitat use and distribution, as well as the identification and spatial trend of the human activities which may directly affect populations traits, is pivotal.

Methods

A MaxEnt modeling approach was applied to predict the seasonal (from April to September) habitat use of a small population of bottlenose dolphins in the north-western Sardinia (Mediterranean Sea) in relation to environmental variables and the likelihoods of boat and fishing net presence. Then, the overlapping areas between dolphin, fishing net and boat presence were identified to provide insights for the marine spatial management of this area.

Results

Three of the main factors influencing the seasonal distribution of bottlenose dolphins in the area are directly (boating and fishing) or indirectly (ocean warming) related to human activities. Furthermore, almost half of the most suitable area for dolphins overlapped with areas used by fishing and boating. Finally, relying on fishing distribution models, we also shed light on the potential impact of fishing on the Posidonia oceanica beds, a protected habitat, which received higher fishing efforts than other habitat types.

Discussion

Modelling the spatial patterns of anthropogenic activities was fundamental to understand the ecological impacts both on cetacean habitat use and protected habitats. A greater research effort is suggested to detect potential changes in dolphin habitat suitability, also in relation to ocean warming, to assess dolphin bycatch and the status of target fish species, and to evaluate sensitive habitats conditions, such as the Posidonia oceanica meadow.

Introduction

Marine coastal biodiversity in the Mediterranean Sea is severely threatened by different anthropogenic pressures (chemical and acoustic pollution, eutrophication, marine traffic, alien species), which are expected to increase in the next future (Coll et al., 2010; Coll et al., 2012). Moreover, the cumulative and synergistic effects of multiple human-induced alterations on coastal ecosystems may impair the attempt to protect marine biodiversity (Micheli et al., 2013).

Among other species, marine mammals in coastal areas suffer habitat fragmentation and loss, reduced availability of resources and disturbance from human recreational activities (Reeves et al., 2003; Reeves, McClellan & Werner, 2013; Natoli et al., 2021). One of the most common species distributed along the Mediterranean coasts in the continental shelf (0–200 m) is the common bottlenose dolphin (Tursiops truncatus) (Bearzi et al., 2008; Gnone et al., 2023), where social units, generally small sized, display limited home ranges and may form isolated and even genetically distinct units at small geographic scales (Gnone et al., 2023; Natoli et al., 2021). The species is strictly protected according to the Habitats Directive (Annex IV; 92/43/ECC), the Protocol for Specially Protected Areas, the Biological Diversity in the Mediterranean Sea of the Barcelona Convention (Annex II), and the ACCOBAMS agreement. Prey depletion, incidental catches in fishing activities and habitat loss are currently the primary threats to bottlenose dolphins in the Basin (Bearzi, Fortuna & Reeves, 2012; Natoli et al., 2021). Further, they may be exposed to high levels of human disturbance, such as shipping, recreational boating, collision, and underwater noise (Bearzi, 2002; Reeves, McClellan & Werner, 2013).

Sustainable management requires spatial mapping of both species distribution and human activities to identify potential risk of conflict. For example, depredation attempts, bycatch and entanglement in fishing gears, and conflict with fishers are most likely to take place where spatial overlap between marine mammal and fisheries occurs (Di Tullio, Fruet & Secchi, 2016), while the risk of collisions and disturbance by boating may be most severe in highly attractive tourist areas where recreational boating is more intense (O’Connor et al., 2009; La Manna et al., 2020). Thus, according to the EU Habitats Directive and the Maritime Spatial Planning Directive of the EU (2014/89), assessing the overlap between dolphins and human activities is crucial to identify conservation strategies and strengthen current policies regarding spatial coastal zone management.

The coasts of Sardinia are considered high-biodiversity areas, where the human impacts can be considered less pronounced than other Mediterranean sites (Coll et al., 2012). However, even in the most pristine coasts, two potentially damaging human activities for dolphins, small-scale fishery and boating, are widely spread. Small-scale fishery (fishing activities performed by vessels 12 m-and-under in length, which do not use towed fishing gears; Davies et al., 2018) is generally considered less impacting on the marine environment than the industrial one. However, it cannot be considered “sustainable” by definition (Davies et al., 2018) due to the potential to over-exploit resources and the mortality to cetaceans because of by-catch (Read, Drinker & Northridge, 2006). Further, conflict with fishers due to the depredation on fishing nets can impair dolphin conservation status (Read, Drinker & Northridge, 2006; Read, 2008) owing to retaliatory measures (intentional injury and killing) taken by fishers. The economic crisis faced by small-scale fishery have exacerbated fishers’ perceptions of dolphins as competitors (Lauriano et al., 2004; La Manna et al., 2022; “Isola Piana: risultati del monitoraggio 2022 e confronto con le annualitá precedenti”; G La Manna, M Moro Merella, A Ruiu, G Ceccherelli, 2022, unpublished data). This is particularly relevant in Sardinia where a depredation frequency of dolphins on gillnets and trammel nets higher than 50% has been recently estimated (La Manna et al., 2022). Sardinia is also one of the Italian regions with the largest concentration of boat berths, tourist ports and marinas (ONN, 2011). Recreational boating is still considered less harmful than shipping for the marine environment, thus it is still scantly regulated (La Manna et al., 2020). However, the impact of boating may lead to dolphin behavioral changes, shifts in habitat use, temporary displacement, increase in energy consumption, and, in the long-terms, changes in survival rates or population size (Constantine, Brunton & Dennis, 2004; Lusseau, 2005; La Manna et al., 2013; La Manna, Manghi & Sarà, 2014).

To promote dolphin conservation, the understanding of their habitat use and distribution, as well as the identification and spatial trend of the human activities which may directly affect populations traits (Carlucci et al., 2016; Lambert et al., 2017; La Manna et al., 2020), is pivotal. Thus, the aim of this study was two folds: (i) to predict bottlenose dolphin habitat suitability in the north-western Sardinia, identifying the environmental and anthropogenic variables that mostly explain the patterns of habitat use; (ii) to estimate the overlaps between dolphin habitat, boating and small-scale fishing activity. The present study updated the fine-scale bottlenose dolphin habitat use previously estimated in the same study area by La Manna et al. (2020), adding the probability of fishing net presence as new predictor. To these aims, a MaxEnt modeling approach (Maximum Entropy; Phillips, Anderson & Schapire, 2006) was used on fishing gear and boat occurrence records to predict the spatial trend of fishing and boating in one of the most popular tourist destinations in Sardinia (Alghero), whose harbor has one of the highest numbers of fishing licenses and boat berths of the island. Then, we used the likelihood of boat and fishing net presence and other environmental variables to predict the dolphin habitat suitability in two seasons (spring and summer 2022) and in the whole period (spring-summer 2022). Finally, the areas of spatial overlap between dolphin, fishing net and boat presence were identified. In fact, if boaters and fishers were mostly distributed in areas highly used by dolphins, then boating and fishing activities could have an effect on dolphin conservation status and habitat conditions which should be better investigated and managed.

Materials & Methods

Study area

The study area (about 400 km2) is in northwest Sardinia, Italy (40.5580 N, 8.3193E, Fig. 1) and includes three coastal protected areas: the Capo Caccia - Isola Piana Marine Protected Area (MPA), the Site of Community Importance (SCI) ‘Capo Caccia e Punta del Giglio’ ITB010042 and the SCI ‘Entroterra e zona costiera tra Bosa, Capo Marargiu e Porto Tangone’ ITB020041. The MPA was established in 2002, mainly for the protection of endemic Mediterranean seagrass (Posidonia oceanica), coralligenous reefs and high priority species, such as the bottlenose dolphin. Human activities (fishing, diving, recreational boating and anchoring) are allowed but regulated in the zones of general protection (Zone B and C), while in Zone A (integral protection) no activity is permitted (with the exception of scientific research authorized by the Management Body). Along the coast the only town is Alghero, whose harbor (the largest on the western coast of the island) hosts 2,200 berths and a fishing fleet mainly composed of small-scale fishing vessels (n = 75; mean length ± SD = 7.18 ± 2.10; “Isola Piana: risultati del monitoraggio 2022 e confronto con le annualitá precedenti”; G La Manna, M Moro Merella, A Ruiu, G Ceccherelli, 2022, unpublished data). The local economy relies on sea-related tourism activities (La Manna et al., 2016); fishing is mainly small scaled and based on traditional practice (“Isola Piana: risultati del monitoraggio 2022 e confronto con le annualitá precedenti”; G La Manna, M Moro Merella, A Ruiu, G Ceccherelli, 2022, unpublished data). Bottlenose dolphin (Tursiops truncatus) is the only species of cetacean regularly present in the area. The local population counts 130 photo-identified individuals, with at least 30% of them showing high site fidelity (La Manna et al., 2023b).

Figure 1 Map of the study area showing the total sampling effort (shades of blue) pooled into grid cells (2 × 2 km). Red points: dolphin sightings between April and September 2022.

Field data collection

Data on fishing, boating and dolphins were collected from April to September 2022. Systematic standardized surveys were conducted as previously described in La Manna et al. (2020) from two identical 9.7 m motorboats, equipped with 270 hp sterndrive engines. Survey routes followed a haphazard sampling procedure (La Manna, Ronchetti & Sarà, 2016; La Manna et al., 2020) and were planned to homogeneously cover the study area with a generally perpendicular direction with respect to the coast and depth contours. In each survey, boat speed was kept between 6 and 10 kn, to ensure dolphin detection, while two experienced observers scanned the sea surface with both naked eyes and with the help of binoculars. Scans occurred from 9 am to 6 pm, only in days with a visibility of over 3 miles and good sea conditions (sea state 2 Douglas; wind force 2 Beaufort). The position of the research boat was automatically recorded every 30 s using a Garmin GPS chart-plotter. Using binoculars (10 power magnification, 42 mm objective lens), trained observers counted boats, both moored and underway, and fishing buoys encountered within 300 m during the survey, recording their position using GPS. Boats were classified as: recreational inflatable boats, speed boats, cabin cruisers, sail boats, diving boats, fishing boats and tourist ferry boats. Fishing buoys (thus the attached gears) were classified as fishing net or traps, based on their peculiar and distinctive positions (Fig. 2). When the classification was ambiguous the gear was classified as not determined (ND) and excluded from the analysis. In case of sighting, dolphins were approached cautiously (to avoid disturbance) within 10–50 m and dolphin group size was estimated by two independent observers. The geographic position of the group (defined as all individuals within visual range that were in apparent association, engaged in the same activity or moving in the same direction; Shane, 1990) was collected using a Garmin GPS chart-plotter at the beginning and at the end of the sighting. Only the beginning sighting points were considered as dolphin occurrence records in the analysis. During each dolphin sighting, a photo-identification data collection was also performed to allow the identification of each dolphin based on the presence of natural marks, (nicks, scars, or skin pigmentations on the dorsal fin;Würsig & Jefferson, 1990). Based on photo-identification analysis, when the same group of individuals was observed several times in the same day, only the first sighting was used in the modelling exercise to avoid the non-independence and temporal autocorrelation of sightings.

Figure 2 Peculiar and distinctive positions of the fishing buoys used to classify the gear as fishing net or traps.

Environmental predictors

We incorporated data on environmental variables, dolphin, fishing gear, and boat occurrence records into a Geographic Information System (GIS; software ArcMap 10.8) using the World Geodetic System 1984 (WGS84) and the Universal Transverse Mercator (UTM) 32 N projection.

The following environmental variables were selected to model dolphin habitat suitability, based on previous studies on cetacean (Azzellino, Panigada & Lanfredi, 2012; Marini, Fossa & Paoli, 2014; Bonizzoni et al., 2019; La Manna, Ronchetti & Sarà, 2016; La Manna et al., 2020): water depth (m), mean sea surface temperature (SSTm), SST range (SSTr, as difference between the maximum and the minimum sea surface temperature), mean chlorophyll concentration (Chl-a - mg/m3), seafloor slope (degree), distance to the coast (m), aspect (downslope direction) and seabed habitat type (Table 1, Suppl. Mat. 2). All data were pooled into grids with a resolution of 250 m and averaged over two periods (spring: from April to June; summer: from July to September) and the whole period (spr-sum). We used ‘Raster Correlations and Summary Statistics’ tools of the plug-in SDMtoolbox 2.5 for ArcGis to create a correlation matrix of the Pearson correlation coefficients (r) between the environmental predictors (Brown, Bennett & French, 2017). Since correlation (r > 0.8) was found between distance to the coast, chlorophyll concentration and depth (Suppl. Mat. 1), only depth was used in the modelling exercises (Kramer-Schadt, Niedballa & Pilgrim, 2013).

Table 1 Description and sources of the predictors used for modelling fishing net, boat and bottlenose dolphin presence.

Predictors	Description and sources	
Water depth	Water depth was mapped using the points recorded every 30 s by the GPS during surveys with very good sea conditions (sea state ≤1). These points were used to create a raster bathymetry surface using “Topo to raster” function of Spatial Analyst tools (resolution 250 m).	
SST (mean and range)	SST source satellite data were downloaded from GOS Project 2010 (https://doi.org/10.5067/GHOUH-4GM20). Monthly climatological raster of GHRSST L4 SST images with 1 km of resolution were created using the plug-in Marine Geospatial Ecology Tools 0.8a68 for ArcGIS. SST raster surfaces were downscaled to the spatial resolution of 250 m using the interpolation tool “Geostatistical Wizard: kernel smoothing” in the Geostatistical Analyst Toolbar and averaged for each season (spring and summer) and for the whole period (spr-sum)	
Chl-a	Chl-a GlobColour monthly L3 products with a resolution of 1 km were downloaded from GlobColour Project (http://globcolour.info). Chl-a raster surfaces were downscaled to the spatial resolution of 250 m using the interpolation tool “Geostatistical Wizard: kernel smoothing” in the Geostatistical Analyst Toolbar and averaged for each season (spring and summer) and for the whole period (spr-sum)	
Slope	Slope defined the bathymetric gradient along the study area and was measured in degrees. A continuous raster surface (250 m resolution) of seabed gradient was derived from water depth using “Slope” function in Spatial Analyst Tools in ArcGIS.	
Aspect	Aspect corresponds to the heterogeneity of the downslope direction and was calculated by deriving the maximum rate of change in water depth values from each cell to its neighbors, using “Aspect” function in Spatial Analyst Tools in ArcGIS. Aspect was a categorical variable, with eight classes, coded as follows: flat (0); N (1); NW (2); E (3); SE (4); S (5); SW (6); W (7); NE (8).	
Distance to the coast	The distance to the coast was calculated for each cell centroid from shoreline shapefile using “Near” function in Spatial Analysis Tools in ArcGIS.	
Distance to the harbor	The distance to harbor was calculated for each raster cell using “Euclidean distance” function of the Spatial Analyst tool. The source to calculate the distance was the closest raster cell to the entrance of the harbor, while the coastline was defined as a barrier. The resulting raster was reclassified using the “Reclassify” function of the Spatial Analyst tool to create a categorical variable with 5 classes of distance coded as follows: 4 (from 0 to 4 km), 8 (from 4 to 8 km), 12 (from 8 to 12 km), 16 (from 12 to 16 km) and 20 (from 16 to 20 km).	
Seabed Habitats	Information about the habitat types were derived from the European Marine Observation Data Network (EMODnet) Seabed Habitats project (http://www.emodnet-seabedhabitats.eu). Habitat type was a categorical variable, with 9 classes, coded as follows: 1 = Mediterranean infralittoral rock (MB15); 2 = Coralligenous biocenosis (MC151); 3 = Mediterranean infralittoral coarse sediment (MB35); 4 = Mediterranean infralittoral sand (MB55); 5 = Biocenosis of Mediterranean muddy detritic bottoms (MC451); 6 = Mediterranean circalittoral coarse sediment (MC35); 7 = Biocenosis of Mediterranean open-sea detritic bottoms on shelf-edge (MD451); 8 = Biocenosis of Posidonia oceanica (MB252); 9= Facies of dead ”mattes” of Posidonia oceanica without much epiflora (MB2523).	

Model generation and validation

We modeled boat, fishing gear, and dolphin occurrence records at different temporal scales by Maximum Entropy (MaxEnt version 3.4.1, Phillips, Dudík & Schapire, 2020), one of the most reliable tools (Elith, Graham & Anderson, 2006; Elith, Phillips & Hastie, 2011) when presence-only data (Guisan & Thuiller, 2005) are available and the number of sightings is low (Elith, Graham & Anderson, 2006; Valavi et al., 2022). MaxEnt estimates a probability distribution (i.e., relative likelihood of presence; McClellan et al., 2014) for a species by contrasting occurrence data with background data, rather than true absence data (Thorne et al., 2012). Thus, MaxEnt does not assume that the absence precludes the likelihood of occurrence, and this aspect is particularly relevant in case of highly elusive species, such as cetaceans (La Manna, Ronchetti & Sarà, 2016; La Manna et al., 2016; La Manna et al., 2020). MaxEnt was preferred to other Species Distribution Model algorithms due to its general higher performance with presence-only data, in comparison with other models (Valavi et al., 2022), and because it allowed to compare the dolphin habitat suitability predicted in 2022 with that relative to the period 2015–2018 obtained with the same modeling approach (La Manna et al., 2020).

Boating and fishing activities were modelled by MaxEnt since they vary temporally and spatially as a function of human behavior (Cummins, Gault & O’Mahony, 2008; La Manna et al., 2020). Thus, we modelled the likelihood of fishing gear and boat presence using the occurrence records of all boats and fishing buoys (distinguished in traps and fishing nets) recorded during the surveys as presence data and the raster surface of depth, slope, aspect, and seabed habitat types as explanatory variables (Table 2; Suppl. Mat. 2). These environmental predictors were chosen since seabed morphology and habitats may affect both fisher and boater behavior. For modeling the likelihood of fishing gear presence, we also considered SSTm and SSTr, since temperature is among the main driver of fish distribution (Azzurro et al., 2019), and distance to harbor (as categorical variable), since the distance between the harbor and the fishing site could influence fishers’ choice (Table 2). Other predictors, such as tidal currents and wave height, that have been used in other fishing modelling exercises (van der Reijden et al., 2018), were not considered here because of their irrelevance in the local (small scaled) context.

Table 2 Percent contribution of the different variables to the MaxEnt models predicting the likelihood of (a) boat (b) fishing net and (c) bottlenose dolphin presence.

Variable importance is presented as the mean of the 10 runs of each single model. Output referred to spring, summer and the whole period (spr-sum).

MaxEnt Model	% Contribution		AUC	AUC difference	
Boat	Depth	Aspect	Slope	Seabed habitat					Mean ± SD	Training - mean test	
Spring	65.3	16.6	16.2	1.9					0.80 ± 0.03	0.017	
Summer	65.7	25.4	5.3	3.7					0.79 ± 0.03	0.007	
Spr-Sum	70.2	21.8	4.4	3.5					0.78 ± 0.02	0.007	
Fishing net	Depth	Aspect	Slope	Seabed habitat	Dist. to harbor	SSTm	SSTr				
Spring	7.9	4.3	4.8	25.2	28.4	28.8	0.5		0.84 ± 0.05	0.020	
Summer	35.7	3.2	1.9	7.4	13.5	14.2	24.1		0.84 ± 0.02	0.021	
Spr-Sum	21.5	2.8	2.3	6.4	17.1	33.9	15.9		0.81 ± 0.02	0.023	
Bottlenose dolphin	Depth	Aspect	Slope	Seabed habitat	SSTm	SSTr	Boat	Fishnet			
Spring	6.1	7.6	4.4	30.3	3.6	2.0	9.9	36.2	0.80 ± 0.08	0.102	
Summer	0.7	16.4	7.9	18.9	11.0	3.6	0.7	40.8	0.78 ± 0.08	0.076	
Spr-Sum	2.8	11.8	9.6	22.9	7.8	1.9	2.5	40.6	0.79 ± 0.05	0.005	

For modeling the likelihood of dolphin presence, we considered four continuous environmental variables (SSTm, SSTr, depth and slope), two categorical environmental variables (aspect and seabed habitat type) and two anthropogenic predictors (the likelihood of boat and fishing net presence modeled by MaxEnt). Since correlation was found between the likelihood of fishing net and trap presence, only the former was used in modelling dolphin presence (Suppl. Mat. 1).

Following La Manna et al. (2020), for all models, MaxEnt settings were chosen in relation to the specific questions of the study and data limitations (Merow, Smith & Silander, 2013): (i) the cloglog output since it is most appropriate for estimating likelihood of presence (Phillips, Dudík & Schapire, 2020), (ii) hinge features (Phillips & Dudík, 2008), (iii) default regularization parameters (0.5), and (iv) 10-fold cross validation to assess the average behavior of the algorithms randomly partitioning the occurrence data in ten random subsets (Phillips, Anderson & Schapire, 2006). The geographical extent of the models coincides with the monitored area that includes the typical and known habitat of bottlenose dolphins (Bearzi, Fortuna & Reeves, 2012). Thus, we used all the background sites available (5,358) to increase the predictive performance (Phillips & Dudík, 2008). Furthermore, since background points were generated from the same environmental space as the presence locations, bias files, useful to fine-tune the selection of background points in MaxEnt and to account for sampling bias (Brown, 2014), were not used.

We ran a jackknife analysis to estimate the contribution of each variable to the MaxEnt run, obtaining alternative estimates of variable importance for our models. The performance of each MaxEnt model was assessed using the AUC (area under the receiver operating characteristic curve; Phillips, Anderson & Schapire, 2006), a threshold-independent metric of overall accuracy. The AUC provides an adequate evaluation of model performance (Phillips, Anderson & Schapire, 2006), even if, in the case of presence-only distribution models, AUC will compare presences with background points and cannot be considered a perfect measure of model accuracy (Lobo, Jiménez-Valverde & Hortal, 2010; Merow, Smith & Silander, 2013). AUC values range between 0 and 1: models with moderate to-good discrimination ability correspond to AUC higher than 0.7 (Swets, 1988). Following a procedure already applied in La Manna et al. (2020), the model robustness was also evaluated calculating (i) the test-AUC standard deviation (SD) and the difference between the train-AUC values (using all presences) and the mean test-AUC values, since lower are the test-AUC SD and the difference between the train-AUC and mean test-AUC values higher is the model robustness (Herkt et al., 2016).

Using the cloglog output of MaxEnt (Phillips, Dudík & Schapire, 2020), we produced maps for each season (spring and summer) and the whole period (spr-sum) relative to the likelihood of boat, fishing net and dolphin presence.

To visualize the overlapping area between dolphins and boats and between dolphins and fishing nets, the intersections of their representative areas were calculated, for both seasons and the whole period, using as representative areas those with likelihood of presence >0.6. This value corresponds to the threshold which balances sensitivity (the change of correctly identifying suitable areas) and specificity (the change of correctly identifying not suitable areas) (Liu, White & Newell, 2013). Finally, two percent area overlap, one between dolphins and boats (PAOd,b) and the other between dolphins and fishing nets (PAOd,f), were calculated following Atwood & Weeks (2003): PAOd,b=Ad,b/Ad×Ad,b/Af0,5

PAOd,f=Ad,f/Ad×Ad,f/Af0,5

where Ad,b isthe overlap area between dolphins and boats, Ad,f isthe overlap area between dolphins and fishing nets, Ad is the representative area for dolphins, Ab is the representative area for boats, and Af is the representative areas for fishing nets.

Results

Boat and fishing net presence

Between April and September, a total of 2,973 boat occurrence records were counted, 657 in spring and 2,316 in summer. The accuracy of MaxEnt in predicting the likelihood of boat presence was higher than 0.75 in any investigated period (Table 2). The most important contributor in predicting the likelihood of boat presence was depth, followed by aspect (Table 2). The probability of finding boats was higher between -60 and -15 m of depth and similar for all aspect classes, except for the classes 2 (NW) and 7 (W), the orientation most exposed to the dominant local wind (Suppl. Mat. 2). Overall, the highest likelihood of boat presence was along the north-western part of the Alghero coast and inside the boundaries of the MPA/SCI (Fig. 3), especially in summer.

Figure 3 Likelihood of boat, fishing net and dolphin presence (cloglog output of MaxEnt models) for spring (upper row), summer (middle row), and the whole period (spring-summer, lower row).

A total of 409 fishing net occurrence records were counted, 175 in spring and 234 in summer. The accuracy of MaxEnt in predicting the likelihood of fishing net presence was higher than 0.8 in any investigated period (Table 2). The most important contributors in predicting the likelihood of fishing net presence were depth (in summer), SSTm, distance to harbor and seabed habitat type (in spring) (Table 2). In spring, the probability of finding fishing nets was higher in the habitat “Mediterranean communities of muddy detritic bottoms” and “Posidonia bed”, and within 4 km to the harbor (Suppl. Mat. 3), while in summer the probability of finding fishing nets was higher between -60 and -20 m of depth and between 8 and 12 km from the harbor. In any period, the increase of SSTm decreased the probability of finding fishing nets. Overall, the likelihood of fishing net presence was highest along the coast closest to Alghero, especially in spring, but also in the areas inside and surrounding the MPA/SCI, especially in summer (Fig. 3).

Dolphin presence

Across all periods, 110 dolphin groups were sighted in about 5,650 km of navigation and 163 surveys. All MaxEnt models obtained AUC higher than 0.75, indicating high accuracy in predicting dolphin presence (Table 2). Further, the overall model robustness can be inferred by the small difference between train-AUC and mean test-AUC values (Table 2).

The most suitable areas for dolphins are inside the Alghero Bay, both along its western and southern coasts, and only partially overlap with the boundaries of the MPA/SCI (Fig. 3). The most important contributing variables to the dolphin habitat suitability models were the likelihood of fishing net presence, seabed habitat type, aspect, SSTm and the likelihood of boat presence (Table 2). Particularly, the probability of finding dolphins: (i) increased as the likelihood of fishing net presence increased, in all seasons; (ii) changed with the seabed habitat types and aspect classes, depending on the season; (iii) decreased when the SSTm increased in summer; iv) decreased when the likelihood of boat presence was higher than 0.5 in spring (Fig. 4).

Figure 4 Effect of SSTm, seabed habitat, likelihood of fishing net presence, likelihood of boat presence, aspect, and slope on the likelihood of dolphin presence (cloglog output) in spring (A), summer (B) and spring-summer (C).

The area with the likelihood of dolphin presence higher than 0.6 extended for about 58 km2, ranging from 49 km2 in spring to 61 km2 in summer (Table 3). This area largely overlapped with both the areas with the highest likelihood of fishing net and boat presence, mainly in summer (Table 3; Fig. 5).

Table 3 Measurement of the overlapping areas between dolphins and fishing nets and between dolphins and boats, in the two seasons (spring and summer) and for the whole period (spr-sum).

Ad: representative area for dolphins; Ab: representative areas for boats; Af: representative areas for fishing nets. PAOdb: percent area overlap between dolphins and boats; PAOdf: percent area overlap between dolphins and fishing nets.

	Spring	Summer	Spr - Sum	
Ad (km)	48.8	60.6	58.0	
Ab (km)	50.7	62.3	70.0	
Af (km)	35.1	39.7	59.7	
PAOdb	17%	43%	41%	
PAOdf	6%	47%	42%	

Figure 5 Overlapping between dolphin and boat representative areas (A) and dolphin and fishing net representative areas (B) in the two seasons (spring and summer) and for the whole period (spr-sum).

PAO: percent area overlap.

Discussion

The study highlighted the spatial patterns of boating and small-scale fishing activity and how they both influence the dolphin habitat suitability.

In particular, based on the MaxEnt outputs, boaters tend to navigate inside the Alghero Bay and the MPA/SCI, the areas most protected by the dominant winds and mostly attractive for the scenic coasts and pristine waters. The most important predictor of boat presence likelihood was depth. The area with the likelihood of boat presence higher than 0.6 extends for 70 km2, ranging from 50 km2 in spring to 62 km2 in summer. These results were consistent with a previous study conducted between 2015 and 2018 (La Manna et al., 2020), and highlighted the constant use by boaters of the same areas over the years. Further, the study measured the extension of the area with the highest likelihood of fishing net presence (about 60 km2, ranging between 35 km2 in spring and 40 km2 in summer) and found the key environmental factors associated with fishing. In spring, fishers tended to use mainly the coastal shallow waters inside the Alghero Bay, at distance to the harbor shorter than 4 km, coherently with the need for small-sized vessels to remain close to the harbor when the local weather conditions are still variable. Fishers showed a clear preference for specific seabed habitat types (namely, muddy detritic bottom and Posidonia oceanica beds), a pattern already observed in other areas (van der Reijden et al., 2018). In summer, fishers operated at greater distance to the harbor (between 8 and 12 km), also using a consistent area within the MPA/SCI. Apart from depth, distance to the harbor and habitat type, SSTm and SSTr were among the most important contributing variables of fishing net presence in both seasons. The fishing hotspots reflect the fishers’ preference for areas with high abundances of the target species, thus the observed decrease of fishnet presence where the SSTm was higher and SSTr lower than 4 °C, could be related to the effect of temperature on fish abundance and distribution.

The modelling exercises on boating and fishing were particularly useful for understanding the influence of the anthropogenic factors on dolphin habitat suitability, coherently with other studies which highlighted the effect of port closeness, shipping routes and fishing areas (Carlucci et al., 2016; Tardin et al., 2019; Maricato, Tardin & Lodi, 2022) on dolphin distribution. The most important contributing variable was the likelihood of fishing net presence, especially in summer, when both fishers and dolphins have used less coastal areas. Consistently, the overlapping area between dolphins and fishing nets in summer reached 47%. The bottlenose dolphin is a species characterized by opportunistic feeding behaviors, known to target a wide range of prey, mostly consisting of demersal fish species (e.g., European hake, red mullet, European conger), benthic fish of soft bottom (e.g., common sole) and cephalopods (e.g., common octopus and common cuttlefish) (Blanco, Salomón & Raga, 2001; Bearzi et al., 2008; Milani et al., 2019), which also represent target species for trammel and gill net fishing. Top predators for which the predation risk is generally low, such as the bottlenose dolphin in the Mediterranean Sea, develop strategies of resources and habitats use based on the optimization of their foraging success (Mannocci et al., 2014). Thus, the present result can be explained by both the depredation behavior of the bottlenose dolphins on fishing nets, a widely spread feeding strategy in the Mediterranean Sea (Reeves, Read & Notarbartolo di Sciara, 2001; Bearzi, 2002; Brotons, Grau & Rendell, 2008; Lauriano et al., 2004; Lauriano et al., 2009; Revuelta et al., 2018; Pardalou & Tsikliras, 2020; La Manna et al., 2022; “Isola Piana: risultati del monitoraggio 2022 e confronto con le annualitá precedenti”; G La Manna, M Moro Merella, A Ruiu, G Ceccherelli, 2022, unpublished data), and the likely sharing of the most productive zones within the study area between dolphins and fishers (Pennino et al., 2015; Carlucci et al., 2016).

Boat presence was a less important predictors, being relevant only in spring, when an increase in the likelihood of boat presence higher than 0.5 leaded to a decrease in the likelihood of dolphin presence. This result seems to contrast with what was found in a previous study in the same area, where boat presence was the most important contributing variable in the dolphin distribution modelling (La Manna et al., 2020). However, this inconsistency could be explained by the shift in the dolphin distribution towards areas less used by boaters, as confirmed by the reduction from 85% (between 2015–2018) to 41% (in 2022) of the overlapping area between dolphins and boats, and by the introduction in the model of a more relevant predictor (the likelihood of fishing net presence), which in turn may be related to prey distribution and dolphin feeding. The latter aspect is also confirmed by the importance of seabed habitat types in modelling dolphin distribution: the preference for certain habitat types, which changes by season, could likely be related to the different prey availability (Hastie et al., 2004; Palacios et al., 2006; Palacios et al., 2013; Gilles, Viquerat & Becker, 2016).

At the end, SSTm also influenced dolphin distribution, mainly in summer, when higher SSTm reduced the likelihood of dolphin presence. The importance of SST in defining dolphin distribution is coherent with recent studies conducted in other geographical areas (Hartel, Constantine & Torres, 2014; La Manna, Ronchetti & Sarà, 2016; Lambert et al., 2017) and with the strong influence of warming and marine heat waves on the occurrence, home range and group size recently found in the same dolphin population (La Manna et al., 2023a; La Manna et al., 2023b). SSTm in 2022 was between 1.19 °C and 1.38 °C higher than the average SSTm recorded in the period 2015–2018 (La Manna et al., 2020). Beside the increased SSTm, also marine heat waves have become longer in the last eight years (La Manna et al., 2023b). Both dolphin and fishing net likelihood of presence were influenced by SSTm and SSTr. Likely, fishers and dolphins respond to the variations in SST adjusting their spatial pattern to the changes in prey populations (Fuller, Mitchell & Maloney, 2016). In fact, ocean warming may impact commercially exploited fish stocks, both directly (acting on fish physiology, behavior, reproduction, and distribution) and indirectly (acting on productivity and structure of ecosystems on which fish depend) (Brander, 2007; Cheung, 2018; Holsman et al., 2020). The changes in the quality, timing, and abundance of fish species may alter the transfer of marine secondary production to higher trophic levels, with cascading implications for food webs (Silber Gregory et al., 2017), and consequent shift in high-level predator distribution and fishing hotspots.

Study limitations

Some ecologically relevant results were obtained with this study, although not all the potential factors influencing dolphin habitat suitability were included in the models. Thus, the limited number of predictors constrains the interpretation of the results within the range of the investigated environmental conditions (La Manna, Ronchetti & Sarà, 2016; La Manna et al., 2016; La Manna et al., 2020). For example, although fishing and SST can be considered good proxies of prey availability, the model predictability would be certainly improved including data on fish abundance and distribution (MacLeod et al., 2014). Further, other aspects remain to be clarified. For example, the distribution of dolphins during the winter months, when sea-based tourist activities cease and fishing reduce due to adverse weather conditions, as well as the influence of different behavioral states (feeding, socializing, resting) on the selection of habitats (Giannoulaki et al., 2017) should be investigated. Finally, considering that marine mammals respond to climate change (Silber Gregory et al., 2017), and early indications on the effect of temperature on the studied population have already been found (La Manna et al., 2023a; La Manna et al., 2023b), further effort should be done to anticipate changes in occurrence, distribution, and relative abundance of this population under climate change scenarios.

Conclusions

Almost half of the most suitable area for dolphins overlapped with areas used by both fishing and boating. Moreover, three of the main factors influencing the seasonal distribution of bottlenose dolphins in the area are directly (boating and fishing) or indirectly (ocean warming) related to human activities, showing the importance of adding anthropogenic activities in the modelling of cetacean habitat use, especially considering the worldwide increase of human pressure on these species (Tardin et al., 2019).

The study confirmed that boating represents a constant anthropogenic pressure in the area, which already showed to influence the distribution pattern of dolphin mother-calf pairs (La Manna et al., 2020) and to change whistling as likely adaptation to the noise produced by boats (La Manna et al., 2019). However, the effect of boating on species’ fitness and conservation status is far to be disentangled (La Manna et al., 2019).

The study provided also the first evidence on the spatial pattern of small-scale fishing activity, whose vessels (less than 12 m in length) are not legally required to carry automatic identification or vessel monitoring systems, given insights into the areas where the chances for dolphins to encounter setnets is higher, as well as the risk of bycatch and entanglement (Breen et al., 2016). No quantitative data regarding the latter aspects exist. However, several cases of survived or stranded dolphins showing signs of entanglements or violent interactions with fishers and dead dolphins due to bycatch were recorded in Sardinia (La Manna, pers. comm, 2023; Fig. 6). Another relevant aspect, from a conservation point of view, is the depletion of fish resources due to intense fishing activities, especially when they are exploited in small-sized areas, as in the present study. The population-level consequences of potential prey depletion for species with varied diet, such as the bottlenose dolphin, are difficult to demonstrate (Natoli et al., 2021). However, the identification of ’hot spot areas’, namely areas with high probabilities of dolphin presence and fishing activities, can lead to more effective protection and management measures (Sofaer et al., 2019), such as the restrictions of fishing in same areas or seasons (Giannoulaki et al., 2017). This aspect is particularly relevant also considering that the local protected areas (MPA and SCIs) are small sized and only slightly overlapped with the areas mostly used by the species. Thus, the present results could be useful for implementing new protected area boundaries and updating current management plans, in order to adjust the local resource exploitation by fishing and minimizing disturbance by boating (La Manna et al., 2020).

Figure 6 Effects of interactions between dolphins and fishery in Sardinia: dolphins showing clear signs of entanglement (A and C) and violent interaction with fishers (who use dive spearguns to get away the animals from the nets, B and D).

Photo credit: Gabriella La Manna.

Further, relying on fishing distribution models, we also shed light on the potential impact of fishing on the Posidonia oceanica beds, which received higher fishing efforts than the other habitat types. Although this is a protected habitat listed in the Habitats Directive (92/43/ECC), to the best of our knowledge, no studies have investigated the ecological impact of fishing on seagrass meadow in this area, either inside the MPA/SCI. Since healthy and functional marine ecosystems depends primarily on the minimization of habitat loss and degradation (Klein et al., 2008), knowledge of the location, extent and ongoing condition of this habitat should be implemented, also to protect the numerous varieties of fish and other marine species seeking food and refuge around the seagrass beds (Waycott, Duarte & Carruthers, 2009), with the aim of preserving both biodiversity and fishing socio-economic viability (Vlachopoulou, Meriwether Wilson & Miliou, 2013).

In conclusion, greater research effort is required to understand the ecological impacts and pressures of boating (Carreño & Lloret, 2021) and fishing, both on dolphins, fish, and protected habitats. In particular, to draw up management strategies aimed to mitigate the identified impacts, the following actions should be implemented: (i) systematic surveys of the bottlenose dolphin population to detect potential changes in distribution patterns, also in relation to climate change; (ii) fishery monitoring to assess dolphin bycatch and the status of target species; (iii) assessment of sensitive habitats conditions, especially that of the Posidonia oceanica meadow.

Supplemental Information

Supplemental Information 1 Dolphin sighting locations used to build MaxEnt models

Click here for additional data file.

Supplemental Information 2 Boat locations used to build MaxEnt models

Click here for additional data file.

Supplemental Information 3 Fishing net locations used to build MaxEnt models

Click here for additional data file.

Supplemental Information 4 Supplementary table and figures

Click here for additional data file.

We want to thank the Capo Caccia—Isola Piana Marine Protected Area, MARETERRA GROUP for logistical support and all students and trainees who helped in the fieldwork. Special thanks to Lauren Polimeno for English editing.

Additional Information and Declarations

Competing Interests

Author Contributions

Data Availability

The authors declare there are no competing interests.

Gabriella La Manna conceived and designed the experiments, performed the experiments, analyzed the data, prepared figures and/or tables, authored or reviewed drafts of the article, and approved the final draft.

Fabio Ronchetti performed the experiments, analyzed the data, prepared figures and/or tables, authored or reviewed drafts of the article, and approved the final draft.

Francesco Perretti performed the experiments, authored or reviewed drafts of the article, and approved the final draft.

Giulia Ceccherelli conceived and designed the experiments, authored or reviewed drafts of the article, and approved the final draft.

The following information was supplied regarding data availability:

The sighting locations used to build the models are available in the Supplementary File.

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
