# Peer review of "Areas of spatial overlap between common bottlenose dolphin, recreational boating, and small-scale fishery: management insights from modelling exercises"

_PeerJ, doi:10.7717/peerj.16111_

## Round 0.1 · original submission · Major Revisions

Dear Authors

The reviewers have commented on your manuscript. You can find attached reports. Based on the comments and suggestions of the expert reviewers, unfortunately, a major revision is needed for your article.

I would like to request you check and correct the manuscript step by step based on the reports.

Sincerely yours

·

Basic reporting

This is an interesting manuscript about how boating and fishing influence bottlenose dolphins habitat suitability in the Mediterranean. It is generally well-written, however, it needs more clarity about i) how it differs from Manna et al (2020), 2) the hypotheses and 3) several aspects of the methodology. Also, authors need to think about of they are really measuring distribution with the methods they are using. Please, see attached file.

Experimental design

The study has a good design, despite it is concentrated in two seasons of 2022. However, several details are missing, for example, what was the GPS sampling protocol. At the analysis section, some parts need more details too (please see attached file), to understand if methods are employed correctly.

Validity of the findings

The findings seems robust and interesting, but should be better explored. Authors may think about exploring more the spatial management aspect related to the MPAs and indicating mitigation strategies for the threat fishing and boating create for bottlenose dolphins, as well as the mismatch between dolphins and protected areas. Authors should also broaden their discussion, comparing with literature outside the Mediterranean and the studied population.

Additional comments

Please, see the attached file.

Reviewer 2 ·

Basic reporting

no comment

Experimental design

line 178: what was the r value in the correlation (0.7-0.8-0.9?)
Why CHL-a was not used in this study. It is a variable that has an important contribution in previous studies (La Manna et al., 2016, 2020).

Validity of the findings

no comment

Reviewer 3 ·

Basic reporting

The overall approach to this study and its design are appropriate for investigating the habitat use of Tursiops truncatus in the study area. In general, the manuscript is well-written, with solid analysis and results.

The manuscript is well-structured and includes adequate tables and figures. Raw data was shared and seems appropriate to the analyses' replicability.

The Introduction brings important background with pertinent literature, although some reduction is recommended.

The authors stated no hypotheses. This is not mandatory in research papers, and I don't believe this jeopardizes the manuscript. The main goals are clear, and let no doubt about the authors' objectives.

Experimental design

This is an updated and complementary study quantifying fine-scale habitat use of common bottlenose dolphin in in northwest Sardinia, Italy, and considers both anthropogenic (probability of fishing nets and boat presence and environmental factors.

The authors adopted a recognized and powerful presence-only SDM tool. However, I recommend some explanations about this method choice, when a presence-absence algorithm could be used once the authors stated they have information about the effort. I am not questioning the choice, but curious (as some of the journal readers probably would be) about it.

On the Methods section, I suggest the author include a description of how they worked on the presence data before the modelling. A description of the separation of the data in training and testing to perform model evaluation is necessary (cross-validation).

Validity of the findings

Although a similar study was performed before in the same area, this manuscript brings new data and improves the modelling by adding one anthropogenic variable not investigated previously. The overall approach to this study and its design are appropriate for investigating habitat use of Tursiops truncatus in the study area.

I suggest the authors to bring the graphical representation of the relationship between the prediction of dolphin presence and the stronger predictors variables to the main text and not the supplementary material.

Figure 3 seems to be incorrect when indicating areas with 100% of probability of dolphin, boat and fishnet occurrence. Please check this up. I never saw a model that predicted 100% of presence probability. Maybe this means Habitat suitability?

In the Discussion section, although all the discussion on the impact of ocean warning on species abundance and distribution is important, here I see lack of evidence to go in this direction. All your data and analyses say is that probability of dolphin sighting and fishnets presence is lower when the temperature was high and the temperature range was low. To advance your discussion on these results to be related to ocean warming, it would be necessary to present information on these variables and how the values included in your modelling represent something higher and lower than expected naturally. If you would like to keep this focus on your discussion, more background on this theme is necessary.

Although the Conclusions address important information, I suggest you review the section. Some of the content seems more appropriate to the Discussion section. Make it straightforward to your main findings.

---

## Round 0.2 · accepted · Accept

I evaluated the revised version of your manuscript. I would like to thank you for considering all reviewer comments and suggestions. I am pleased to inform you that your article has been accepted
Sincerely yours

Reviewer 2 ·

Basic reporting

The authors have made the necessary improvements.

Experimental design

The authors have made the necessary improvements.

Validity of the findings

The authors have made the necessary improvements.

Additional comments

The authors have made the improvements requested by the referees. It is suitable for publication in this form.
best regards

Reviewer 3 ·

Basic reporting

No comments

Experimental design

No comments

Validity of the findings

NO comments

Additional comments

Thank you for your revision. Most of my suggestions were considered or adequately replied. I have only minor suggestions (see the attached document). However, I am completely satisfied with the new version of the manuscript. Congratulations on your research, and I hope your results can be applied in practice for the species conservation.

Annotated reviews are not available for download in order to protect the identity of reviewers who chose to remain anonymous.